# Pneumoconiosis with a Sarcoid-Like Reaction Other than Beryllium Exposure: A Case Report and Literature Review

**DOI:** 10.3390/medicina56110630

**Published:** 2020-11-22

**Authors:** Fumiko Hayashi, Takashi Kido, Noriho Sakamoto, Yoshiaki Zaizen, Mutsumi Ozasa, Mitsuru Yokoyama, Hirokazu Yura, Atsuko Hara, Hiroshi Ishimoto, Hiroyuki Yamaguchi, Taiga Miyazaki, Yasushi Obase, Yuji Ishimatsu, Yoshinobu Eishi, Junya Fukuoka, Hiroshi Mukae

**Affiliations:** 1Department of Respiratory Medicine, Nagasaki University Graduate School of Biomedical Sciences, 1-7-1 Sakamoto, Nagasaki 852-8501, Japan; fumi.h369@gmail.com (F.H.); nsakamot@nagasaki-u.ac.jp (N.S.); 0717mutumi@gmail.com (M.O.); h_yura.5994@me.com (H.Y.); a-hara@nagasaki-u.ac.jp (A.H.); h-ishimoto@nagasaki-u.ac.jp (H.I.); yamaguchi-hiroyuki@umin.ac.jp (H.Y.); obaseya@nagasaki-u.ac.jp (Y.O.); hmukae@nagasaki-u.ac.jp (H.M.); 2Department of Pathology, Nagasaki University Graduate School of Biomedical Sciences, 1-7-1 Sakamoto, Nagasaki 852-8520, Japan; zaizen_yoshiaki@med.kurume-u.ac.jp (Y.Z.); fukuokaj@nagasaki-u.ac.jp (J.F.); 3Department of Anatomy, University of Occupational and Environmental Health, 1-1, Iseigaoka, Yahatanishiku, Kitakyushu City, Fukuoka 807-8555, Japan; spacehunter7000@gmail.com; 4Department of Infectious Diseases, Nagasaki University Graduate School of Biomedical Sciences, 1-7-1 Sakamoto, Nagasaki 852-8501, Japan; taiga-m@nagasaki-u.ac.jp; 5Department of Nursing, Nagasaki University Graduate School of Biomedical Sciences, 1-7-1 Sakamoto, Nagasaki 852-8520, Japan; yuji-i@nagasaki-u.ac.jp; 6Department of Human Pathology, Tokyo Medical and Dental University, Yushima, Bunkyo-ku, Tokyo 113-8519, Japan; eishi.yoshi@gmail.com

**Keywords:** aluminum, berylliosis, chronic beryllium disease, pneumoconiosis, sarcoidosis, sarcoid-like reaction, X-ray analytical electron microscopy

## Abstract

*Background:* Chronic beryllium disease (CBD) is a granulomatous disease that resembles sarcoidosis but is caused by beryllium. Clinical manifestations similar to those observed in CBD have occasionally been reported in exposure to dusts of other metals. However, reports describing the clinical, radiographic, and pathological findings in conditions other than beryllium-induced granulomatous lung diseases, and detailed information on mineralogical analyses of metal dusts, are limited. *Case presentation:* A 51-year-old Japanese man with rapidly progressing nodular shadows on chest radiography, and a 10-year occupation history of underground construction without beryllium exposure, was referred to our hospital. High-resolution computed tomography showed well-defined multiple centrilobular and perilobular nodules, and thickening of the intralobular septa in the middle and lower zones of both lungs. No extrathoracic manifestations were observed. Pathologically, the lung specimens showed 5–12 mm nodules with dust deposition and several non-necrotizing granulomas along the lymphatic routes. X-ray analytical electron microscopy of the same specimens revealed aluminum, iron, titanium, and silica deposition in the lung tissues. The patient stopped smoking and changed his occupation to avoid further dust exposure; the chest radiography shadows decreased 5 years later. *Conclusion:* The radiological appearances of CBD and sarcoidosis are similar, although mediastinal or hilar lymphadenopathy is less common in CBD and is usually seen in the presence of parenchymal opacities. Extrathoracic manifestations are also rare. Despite limited evidence, these findings are similar to those observed in pneumoconiosis with a sarcoid-like reaction due to exposure to dust other than of beryllium. Aluminum is frequently detected in patients with pneumoconiosis with a sarcoid-like reaction and is listed as an inorganic agent in the etiology of sarcoidosis. It was also detected in our patient and may have contributed to the etiology. Additionally, our case suggests that cessation of dust exposure may contribute to improvement under the aforementioned conditions.

## 1. Introduction

Studies have shown that exposure to industrial dust may lead to the development of granulomatous lung disease. Hardy and Tabershaw have described a granulomatous disease that resembles sarcoidosis but is caused by beryllium, known as berylliosis or chronic beryllium disease (CBD) [1]. Clinical manifestations similar to those observed in CBD have occasionally been reported in cases of exposure to dust of other metals [2,3,4,5,6,7,8,9,10,11,12,13]. However, reports describing the clinical, radiographic, and pathological findings of conditions other than beryllium-induced pneumoconiosis with a sarcoid-like reaction, and detailed information on mineralogical analyses of metal dusts are limited. Herein, we report a case of pneumoconiosis with a sarcoid-like reaction other than CBD and review similar reported cases in the literature.

## 2. Case Presentation

A 51-year-old, asymptomatic Japanese man with rapidly progressing nodular shadows, as observed on chest radiography performed as part of his routine physical examinations, was referred to our hospital. Two years ago, his chest radiograph had revealed no abnormalities. Small nodules had appeared a year later, mainly in both the lower lung fields (Figure 1A); however, he did not visit a hospital. Further, he reported a smoking history of 30 pack-years. As a civil engineer, he had been engaged in underground construction with welding, cutting, and drilling of reinforced concrete and metals for 10 years (between the ages of 41 and 51 years); though he had never used and/or been exposed to beryllium.

The patient did not display clinical respiratory symptoms such as dyspnea, exercise intolerance, or any other relevant manifestations upon referral to our hospital. The patient’s body temperature, pulse rate, and respiratory rate were 36.9 °C, 58 beats/min, and 20 breaths/min, respectively. On chest auscultation, fine crackles were audible in the bilateral lower lung fields. Clubbed fingers were not observed. Laboratory data on admission revealed elevated levels of PR3-ANCA (15.4 U/mL, cutoff level: 3.5 U/mL). Other representative tests for collagen disease-related autoantibodies were negative. Serum levels of angiotensin-converting enzyme, carcinoembryonic antigen, KL-6, and sIL-2R were in the normal range, and the results of interferon-gamma release assay were negative. The chest radiograph on admission showed an increase in the number of nodules compared with those found 1 year previously (Figure 1B).

High-resolution computed tomography (HRCT) (Figure 2) showed well-defined multiple centrilobular and perilobular nodules, and thickening of the intralobular septa in the middle and lower zones of both lungs. No pleural effusion or mediastinal lymphadenopathy was observed.

Positron emission tomography showed no significant accumulation on the nodules. No extrathoracic manifestations were observed, both clinically and radiologically. The pulmonary function test results revealed the following: total lung capacity, 4.14 L (116.3% of the predicted value); forced vital capacity, 4.00 L (112.4% of the predicted value); forced expiratory volume in 1 s, 3.16 L (109.0% of the predicted value); and carbon monoxide diffusion capacity of the lung, 6.55 mL/min/mm Hg (88.2% of the predicted value). An analysis of the bronchoalveolar lavage fluid obtained from the middle lobe (right B^5^) showed 2.4 × 10^5^ cells/mL (88.9% macrophages, 9.9% lymphocytes, 1.0% neutrophils, and 0.0% eosinophils), with a CD4/CD8 ratio of 0.7. Lung specimens from the anterior basal segment of the right lower lobe, obtained by transbronchial lung biopsy, exhibited non-necrotizing granulomatous inflammation and chronic multinucleated giant cells with marked dust deposition. To confirm the diagnosis, lung specimens from the lateral segment of the right middle lobe and anterior basal segment of the right lower lobe were obtained via video-assisted thoracoscopic surgery. The specimens showed 5–12 mm silicotic nodules with dust deposition (silica- and berlin blue staining-positive dust), surrounded by fibrotic changes and macrophages on the pleura and around the airways (Figure 3A,B). Hemosiderin-laden macrophages were observed in the surrounding alveoli of the nodules. Several non-necrotizing granulomas along the lymphatic routes were also observed (Figure 3C,D). No asbestos bodies were observed, and immunostaining for *Cutibacterium acnes*, Ziehl–Neelsen staining, and Grocott staining were all negative.

X-ray analytical electron microscopy (S-4500 Hitachi, Japan) of the same specimens detected particles containing aluminum, iron, titanium, and silica (Figure 4).

Based on his history of occupational exposure and his clinical, radiological, and pathological findings, the patient was diagnosed with pneumoconiosis with a sarcoid-like reaction. He did not receive any medical treatment, such as corticosteroid administration; however, he stopped smoking and changed his occupation to avoid further exposure to dust. Two years later, the HRCT showed an increase in the nodular shadows (Figure 5A). However, the shadows were observed to have decreased after 5 years (Figure 5B).

## 3. Discussion

Non-necrotizing granulomas are observed in several lung diseases, including sarcoidosis, and their causes can be broadly classified into infectious and noninfectious factors. Noninfectious factors include the inhalation of organic and inorganic agents [14,15]. CBD is a well-known type of pneumoconiosis with a sarcoid-like reaction, which is caused by beryllium exposure, and shares several clinical and histopathological features with sarcoidosis [1,16,17]. Similar manifestations have occasionally been reported in cases of exposure to dust of other metals, such as aluminum, titanium, silica, and zirconium [2,3,4,5,6,7,8,9,10,11,12,13]. Our patient showed non-necrotizing granulomatous inflammation with chronic-multinucleated giant cells and marked dust deposition without a history of beryllium exposure; additionally, aluminum, iron, titanium, and silica were detected in the deposited dust in the lung specimens. Therefore, we suspected the development of pneumoconiosis with a sarcoid-like reaction in our patient due to exposure to occupational dust of metals other than beryllium.

The appearance of CBD in the radiological findings is similar to that of sarcoidosis, although mediastinal or hilar lymphadenopathy is less common and is usually seen in the presence of parenchymal opacities [18,19]. On chest CT, multiple nodules are the most common finding, often clustered around the bronchi and interlobular septa or in the subpleural region. Ground-glass opacities, bronchial wall thickening, and interlobular septum thickening may also be seen [18,19]. Compared with patients with sarcoidosis, extrathoracic manifestations, such as those observed in the eye, skin, heart, liver, and nervous system are rare in patients with CBD [18]. Further, there is little evidence of pneumoconiosis with a sarcoid-like reaction due to exposure to dust other than beryllium dust; however, similar findings have been reported in four out of five cases based on detailed clinical and radiological information [4,8,9,10,13]. Multiple nodules have also been observed in these four out of five cases [4,8,9,10,13], and mediastinal and hilar lymphadenopathy was observed in only one case [8]. Extrathoracic manifestations have not been observed. In our case, multiple nodules and thickening of the intralobular septa were the main radiological findings, and lymphadenopathy and extrathoracic manifestations were not observed. These findings correspond to pneumoconiosis with a sarcoid-like reaction.

The natural history of CBD has been variable, and the duration from beryllium sensitization to CBD development is reportedly 3.8 years (range, 1.0–9.5 years) [20]. The clinical course varies, and most patients experience a gradual deterioration in their condition; however, improvement may occur with the cessation of beryllium exposure [18,20,21]. Besides beryllium exposure, 18 months of toner dust exposure and 3 years of magnetite iron exposure have been shown to cause pneumoconiosis with a sarcoid-like reaction, without improvement in radiological findings despite administration of corticosteroid therapy [8,13]. By contrast, radiological findings of another individual exposed to aluminum dust for 15 years improved with corticosteroid therapy [4]. Further, there was no improvement in radiological findings in an individual exposed to glass wool fibers for 7 years, even after 4 years of exposure cessation [10]. However, in a case of pneumoconiosis with a sarcoid-like reaction in a limestone quarry worker with 8 years of employment, the radiological findings resolved 2 years after the relocation of his workplace [9]. In our case, the patient was engaged in underground construction as a civil engineer, with welding, cutting, and drilling of reinforced concrete and metals for 10 years, and the rapid progression of nodular shadows was observed over 2 years. Improvement in his condition was noted 5 years after exposure cessation. Due to limited evidence for pneumoconiosis with a sarcoid-like reaction, it is difficult to discuss the differences in the clinical course of the patients, and the effect of exposure cessation on the resolution of radiological findings. However, our findings largely suggest that cessation of exposure may improve progression of the disease [9].

In this case, X-ray analytical electron microscopy detected aluminum, iron, titanium, and silica deposition in the lung tissues. In four case reports of pneumoconiosis with a sarcoid-like reaction other than CBD, detailed mineralogical analyses were performed and the following were detected in the lung tissues: aluminum and silica due to limestone exposure; aluminum, silica, and magnesium due to glass wool fiber exposure; silica and iron due to toner dust exposure; and aluminum, magnesium, and silica due to metal reclamation factory exposure [4,8,9,10,13]. Additionally, aluminum, silica, and titanium were detected in the lung tissues of six patients with pneumoconiosis with a sarcoid-like reaction due to man-made mineral fiber exposure [2]. Other studies have also shown that aluminum and silica are frequently detected in patients with pneumoconiosis with a sarcoid-like reaction [3,7]. We cannot ignore the effects of silica, as it is detected in most lung samples of various lung diseases, regardless of the extent of occupational exposure [22,23].

Aluminum is frequently detected in patients with occupational exposure [22,23] and can cause pulmonary fibrosis, referred to as aluminosis or aluminum lung [24,25]. Pathological findings include interstitial fibrosis and, occasionally, granuloma [24,26]. Aluminum has also been listed as an inorganic agent in the etiology of sarcoidosis [14]. Collectively, we speculate that aluminum may play a particularly important role in the pathology of pneumoconiosis with a sarcoid-like reaction.

CBD is diagnosed using the beryllium lymphocyte proliferation test [18]. A recent study has highlighted the utility of the lymphocyte proliferation test called memory lymphocyte immunostimulation assay (MELISA^®^) in identifying metals causing sarcoid-like reactions; in this test, 9 out of 13 patients (69%) tested positive for at least one of the metal components [27]. However, as this method is not routinely used in Japan and our patient showed a favorable response to avoiding dust exposure, we did not perform this test. Although the specificity of this technique for pneumoconiosis with a sarcoid-like reaction is unclear, it may be used to confirm the pathology or to identify the underlying etiology, particularly in cases with an unfavorable response.

The detection of mineral components using mineralogical analyses, the beryllium lymphocyte proliferation test, and/or MELISA^®^ is not a convenient option. However, these tests are recommended in all patients with sarcoidosis or sarcoid-like reaction. Based on the presentation of our case, we suggest that these tests may be beneficial for exploring the cause of the condition and the effects of cessation of exposure in patients with a history of occupational exposure, and/or dust deposition in lung tissues.

Recently, beryllium sensitization caused by environmental exposure to beryllium-containing concrete dust in a cluster of workers from a non-beryllium-related industry was reported in Germany [28]. While our patient did not have a history of beryllium exposure, we cannot exclude the possibility of CBD in the present case, in addition to the previous reports of pneumoconiosis with a sarcoid-like reaction without beryllium exposure [4,8,9,10,13]. Further expansion of methods for the diagnosis of CBD is also required.

## 4. Conclusions

The radiological findings of CBD are similar to those of sarcoidosis, although mediastinal or hilar lymphadenopathy is less common and is usually seen in the presence of parenchymal opacities. Extrathoracic manifestations are rare in patients with CBD. Moreover, besides CBD, the findings observed in pneumoconiosis with a sarcoid-like reaction may be similar to those observed in other dust exposure cases. The findings in our case were compatible with those observed in pneumoconiosis with a sarcoid-like reaction, and suggest that this condition may be improved with the cessation of dust exposure. Aluminum was also detected in this case and may have played an important role in the clinical manifestations. As the information on pneumoconiosis with a sarcoid-like reaction due to exposure to dust other than beryllium is limited, further research is warranted for appropriate diagnosis and management.

## Figures and Tables

**Figure 1 medicina-56-00630-f001:**
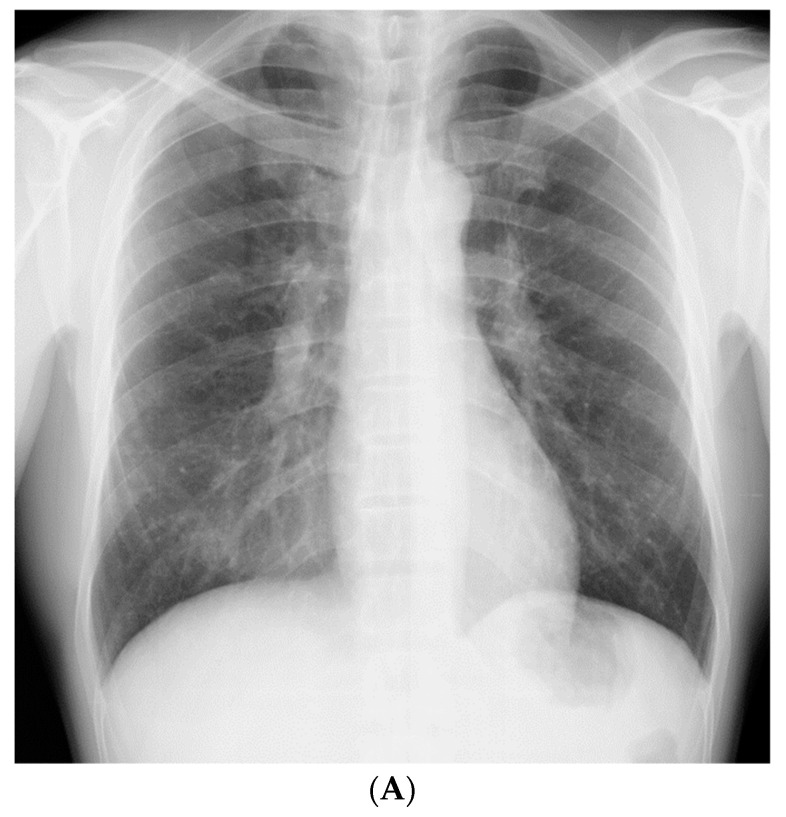
(**A**) Chest radiograph 1 year prior to referral to our hospital, showing small nodules in both the lower lung fields. (**B**) Chest radiograph on admission, showing an increase in the number of nodules in 1 year.

**Figure 2 medicina-56-00630-f002:**
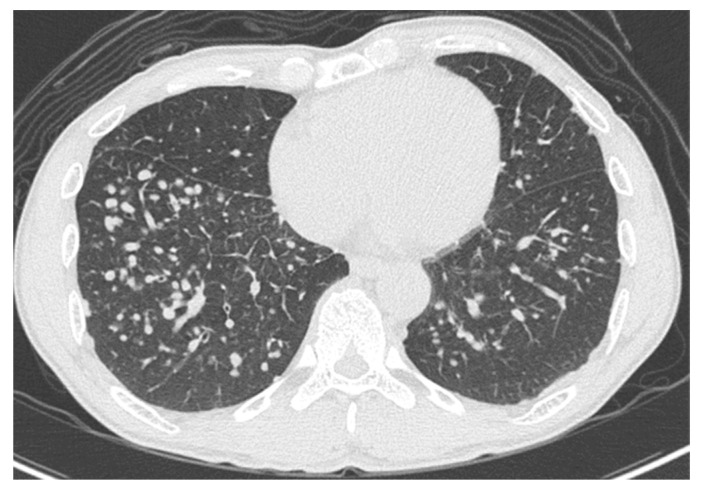
High-resolution computed tomography of the lower lung lobes on admission, showing well-defined multiple centrilobular and perilobular nodules, and thickening of the intralobular septa in the middle and lower zones of both lungs.

**Figure 3 medicina-56-00630-f003:**
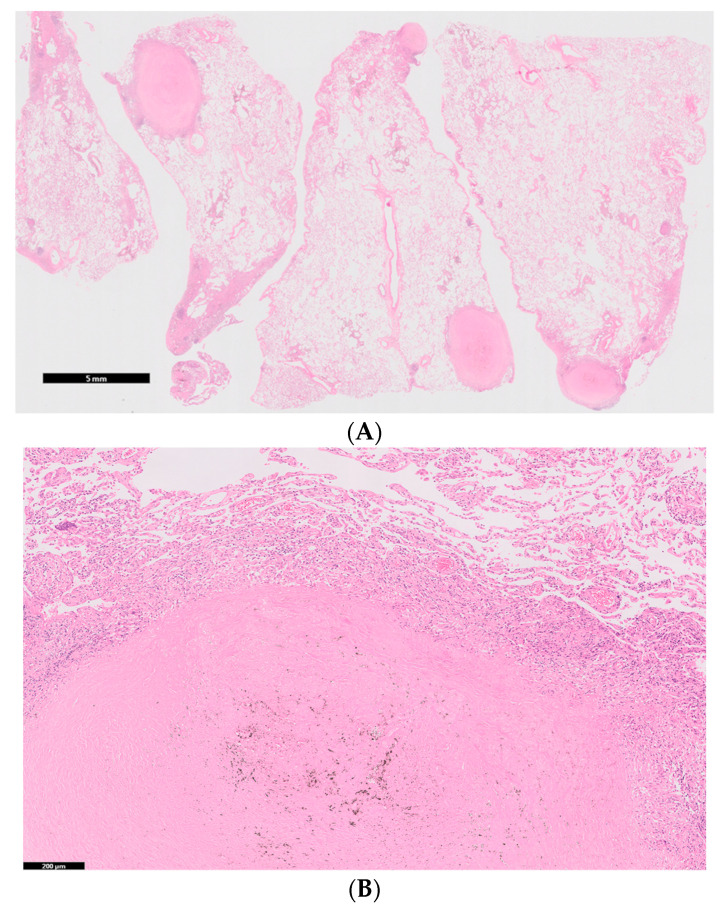
(**A**). Lung specimens from the lateral segment of the right middle lobe and anterior basal segment of the right lower lobe, obtained via video-assisted thoracoscopic surgery showing 5–12 mm sized nodules with dust deposition. The length of the scale bar is 5 mm; (**B**). Higher magnification of the nodules with dust deposition, surrounded by fibrotic changes and macrophages. The length of the scale bar is 200 μm; (**C**). Non-necrotizing granulomas along the lymphatic routes. The length of the scale bar is 500 μm; (**D**). Higher magnification of the non-necrotizing granulomas. The length of the scale bar is 100 μm.

**Figure 4 medicina-56-00630-f004:**
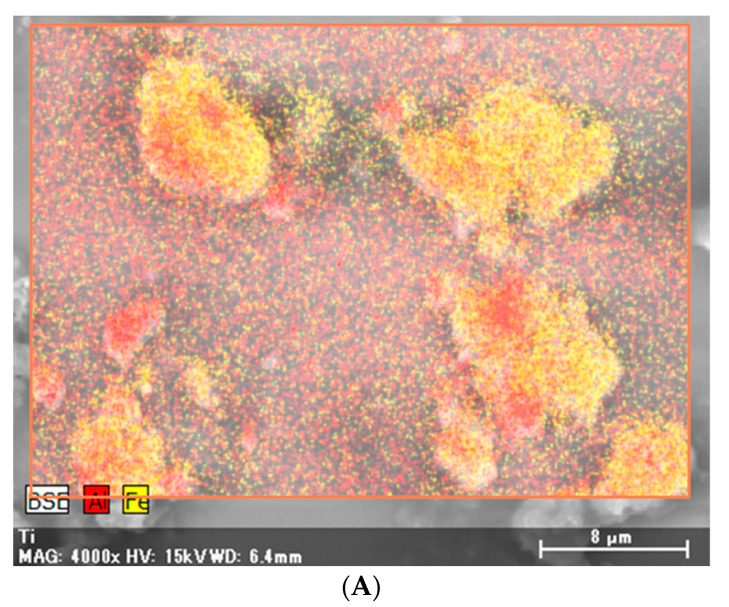
X-ray microanalysis of particles in the lung specimens. Particles containing aluminum (Al) (**A**–**C**), iron (Fe) (**A**,**B**,**D**), titanium (Ti) (**E**,**F**), and silica. In Figure 4B,F, background peaks are also observed.

**Figure 5 medicina-56-00630-f005:**
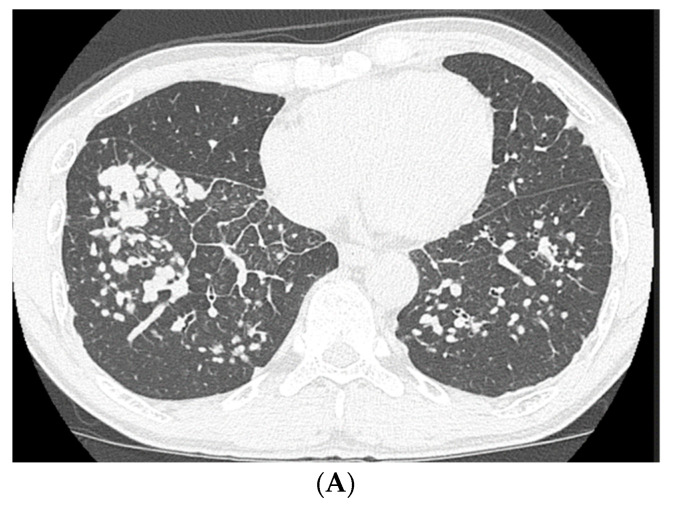
**(A**) High-resolution computed tomography of the chest after 2 years, showing an increase in the nodular shadows. (**B**) High-resolution computed tomography (HRCT) of the chest after 5 years, showing a decrease in nodular shadows.

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
