# Peer review of "Pneumoconiosis with a Sarcoid-Like Reaction Other than Beryllium Exposure: A Case Report and Literature Review"

_medicina, 2020, doi:10.3390/medicina56110630_

Round 1

Reviewer 1 Report

Hayashi and co-authors have presented a case of a granulomatous lung disease potentially due to occupational exposures. While the exposures described in this case are not novel, it is an interesting observation given the multiple potential etiologies discovered and the literature review is timely and warranted.

Comments:

  1. I like the stated diagnosis of "pneumoconiosis with a sarcoid-like reaction" and I think this is an appropriate description that confers important information and should be adopted more widely in the literature.
  2. Given the recent report of beryllium exposure in individuals working with concrete (Frye, CHEST 2020), please comment on the likelihood that this patient may have had unsuspected chronic beryllium disease.
  3. The pathologic features of figure 3A are atypical for either sarcoidosis or chronic beryllium disease. These appear to be more consistent with silicotic nodules, or perhaps nodular sarcoidosis. It is difficult at this low-power to see whether there are granulomas surrounding the central hypocellular areas. A higher power view of the nodule would be helpful.  At this power, the features are unlikely to be confused with sarcoidosis. It would be useful to have these images reviewed by a pulmonary pathologist.
  4. It is difficult to appreciate that the granuloma depicted in figure 3C is in a lymphatic distribution. A lower power view would be necessary for that.
  5. It may be useful to the reader if the authors can provide some comments on which patients should undergo polarization, and which patient should undergo mass spectroscopy when a diagnosis of sarcoidosis is being entertained? Given that this is a difficult test to perform, can the authors suggest which patient should be referred for it?

Minor comments:

  1. Berylliosis is an outdated label for chronic beryllium disease
  2. S4, S8 etc are not commonly used. Consider changing these to lateral segment of the right middle lobe anterior basal segment of the right lower lobe etc.

Reviewer 2 Report

The authors describe a case report of pneumoconiosis with sarcoid-like reaction other than beryllium exposure and present a review of the literature describing this phenomenon. In this particular case, the patient developed worsening lung nodules which resolved after exposure reduction and smoking cessation.

Overall this is a well written manuscript with excellent quality images and figures enhancing the readability. I have minor revisions as outlined below:

  1. In the manuscript it states that the patient was referred because of 'rapidly progressing nodular shadows' on CXR. Did the patient have any clinical symptoms of dyspnea, exercise intolerance or any other clinical respiratory symptoms?
  2. Could the authors expand on the clinical course of the patient while in hospital as part of the case report section? How long was the patient admitted? If there were clinical symptoms on admission, did they resolve?
  3. Is there any follow-up PFT or ANCA measurements other than those performed on admission to see if these resolved and correlated to with the nodule resolution on CT at 5-years?
  4. Was the patient treated with any medical treatment while in hospital, such as corticosteroid? Was this treatment continued after discharge?
